# Automated Error Labeling in Radiation Oncology via Statistical Natural Language Processing

**DOI:** 10.3390/diagnostics13071215

**Published:** 2023-03-23

**Authors:** Indrila Ganguly, Graham Buhrman, Ed Kline, Seong K. Mun, Srijan Sengupta

**Affiliations:** 1Department of Statistics, North Carolina State University, Raleigh, NC 27607, USA; 2Department of Educational Psychology, University of Wisconsin–Madison, Madison, WI 53706, USA; 3RadPhysics Services LLC, Albuquerque, NM 87111, USA; 4Arlington Innovation Center, Health Research, Virginia Tech, Arlington, VA 22203, USA

**Keywords:** patient safety, medical errors, neural networks, text classification, statistical modeling

## Abstract

A report published in 2000 from the Institute of Medicine revealed that medical errors were a leading cause of patient deaths, and urged the development of error detection and reporting systems. The field of radiation oncology is particularly vulnerable to these errors due to its highly complex process workflow, the large number of interactions among various systems, devices, and medical personnel, as well as the extensive preparation and treatment delivery steps. Natural language processing (NLP)-aided statistical algorithms have the potential to significantly improve the discovery and reporting of these medical errors by relieving human reporters of the burden of event type categorization and creating an automated, streamlined system for error incidents. In this paper, we demonstrate text-classification models developed with clinical data from a full service radiation oncology center (test center) that can predict the broad level and first level category of an error given a free-text description of the error. All but one of the resulting models had an excellent performance as quantified by several metrics. The results also suggest that more development and more extensive training data would further improve future results.

## 1. Introduction

A milestone report from the Institute of Medicine (IOM), published in 2000, brought public and political attention to the severe fallout of medical errors and highlighted the need to address medical errors and their effects on human health [1]. More recently, medical errors have been shown to be the third leading cause of death in the United States [2]. Based on quality-adjusted life years, the number of preventable deaths caused by medical mistakes is now 10 times higher than what the IOM estimated [2]. One study estimated that preventable medical errors may cost the U.S. economy up to $1 trillion in lost human potential and contributions [2]. Although there has been an intense focus over the past two decades to improve the safety of medicine in the U.S., progress has been slow.

One of the challenges of addressing medical errors is the manner in which errors are reported and the sheer amount of information provided in such reports [3]. Medical errors have traditionally been reported as free-text descriptions by front-line staff [4]. Inconsistencies occur as reporters use different levels of detail and various types of terminology and vocabulary for both technical and generic uses of language. Furthermore, one of the primary barriers to error reporting is the varying perspective on errors themselves, since reporters may have conflicting perspectives regarding the incident being reported [5]. Given these problems associated with the analysis of error reports, natural language processing (NLP), the branch of machine learning that deals with human language and speech, has emerged as a potential technology in patient safety research to improve error prevention through better detection, reporting, and analysis [4,6,7,8]. A significant benefit of NLP models for medical personnel is the reduction in time and effort required to report medical errors. These predictive models make the process easier by advising reporters on the possible categorization of an error report based on statistical confidence. This allows reporters to make more reliable and timely decisions upstream, before the error propagates downstream through the various clinical pathways. Furthermore, the model’s suggestions remove heuristic bias from error reports because they are based on patterns observed across all reports, not just the experiences of one staff member [9].

Radiation therapy involves a complex process workflow composed of as many as 300 steps and many interactions among various systems, devices, and medical personnel (see Figure 1). Because of this, the field of radiation oncology is particularly prone to errors. Any single error or combination of errors can have an adverse measurable impact on patient outcomes. Ineffective error management can lead to reduced quality, increased inefficiency, and increased legal and/or regulatory liabilities [10,11,12].

Research on using NLP for medical errors in radiation oncology has received limited attention, although recent work shows promise in developing models that are effective in error labeling/classification [13,14]. Previous work has been focused on safety assurance and error reduction, but few studies have incorporated NLP into their methodology. These studies have, however, identified trends and means by which errors can be effectively uncovered and reported, providing a foundation for our modelling work in this study.

Our research examines the use of NLP models as a potential mechanism for error reporting, analysis, and modeling of error propagation in radiation oncology. We collected data from the clinical database of a radiation oncology treatment center (“test center”) and developed several multi-class classification models to predict the error categories of reports. The model development goals were to find a way to effectively use an unevenly distributed data set for the training of multi-class classification models, and to accurately categorize errors from a highly heterogeneous and noisy set of free text reports submitted by personnel at the test center. Predicting when and where errors occur throughout the treatment delivery process could provide a better systematic understanding of risk management in patient safety and quality of care.

One of the best practices identified in prior literature is the use of input and recommendations received from the entire radiation oncology team (e.g., radiation oncologist, medical physicist, dosimetrist, linac engineer, therapist, nurse, financial counselor/biller, receptionist, and management representative) in the model development process [15]. Following this practice, we chose to build upon deep and diverse data provided by an established, commercially available error reduction and regulatory compliance program called Medical Error Reduction Program (MERP©) (https://www.radphysics.com accessed on 15 January 2023). A product of RadPhysics Services LLC (RPS), MERP provides a comprehensive framework for input and processing of preventable systems-related errors by members of the radiation oncology team. RPS supplied the test center data. Another theme seen in clinical practice and supported by various professional organizations is the identification of various check points (e.g., dose calculation checks, data entry checks, chart checks, etc.) in the treatment process that serve as places to detect errors [16]. We have incorporated this into our framework to keep a common language, which ensures that other researchers and practitioners will be able to understand the results and apply them to their future work.

## 2. Data Description and Problem Statement

The data collected from the test center’s clinical database consisted of error reports from both pre-treatment errors (errors discovered before treatment began) and post-treatment errors (errors caught after treatment started). We are interested in two fields: the description field, which contains the free text report of the incident, and the event category field, which contains the broad category of error, i.e., in which step of the clinical workflow did the error occur. Our data also included reports of quality assurance, radiation safety, and medical billing errors. In current practice, determining the category or label of each error incident from available data requires a qualified individual. This manual labeling task requires dedicated time and resources and is subject to the reporter’s heuristic interpretation. The goal of our research was to create a statistical model that can automate this labeling task by taking the free-text narrative as input and predicting the event category as output.

There are n=1121 reports, and the event category field had 16 different labels. As seen in Figure 2, the event categories are heavily imbalanced in frequency, and several categories are too small for statistical modeling. Therefore, we grouped the 16 event categories into four broad categories: Administrative, Standards, Treatment, and Treatment preparation. The broad categories comprise a detailed breakdown of the overall patient treatment process. These categories are defined as follows: (1) “Administrative” encompasses patient data such as patient information, demographics, reports, etc., (2) “Standards” entails compliance items such as billing, QA and radiation safety recommendations and requirements for various machine, equipment, measurements, checks, etc., (3) “Treatment” means the steps in the administration of radiation including patient setup, imaging, treatment delivery, etc., and (4) “Treatment Preparation” comprises the overall groundwork for initiating patient treatment such as diagnosis/staging, simulation, prescription, treatment planning, schedule, etc. Figure 2 shows the frequency distribution of the 16 event categories (left) and the frequency distribution of the four broad categories (right) is color-coded to show the assignment of event categories into broad categories. For example, the orange color used for the event categories Scheduling, Registration, and Patient Docs/Notes is also assigned to the broad category Administrative.

## 3. Materials and Methods

We first considered the problem of classifying the full set of reports into the four broad categories. Next, we separately considered the set of reports under each broad category and considered the problem of classifying them into the corresponding event categories. Following the standard supervised learning workflow, we first constructed the training data by randomly extracting a subset of rows from the full dataset, and then used the training data to fit a classification model. The fitted model was then applied to the test data, which consists of the data points that were not selected in the training data. To quantify model performance, the predicted labels for the test data were compared to the reported labels. The training proportion was set at 70%.

For the classification task to be meaningful, the test data must include at least one report from each category. For a given category, suppose there are *n* reports from that category in the full dataset, and let *X* be the number of reports from that category included in the test data. Since we randomly selected 30% of the reports to be included in the test data, *X* follows a Hypergeometric distribution with parameters *N*, the population size (i.e., the total number of reports), and *n*, the group size (i.e., the number of reports from that particular category). Therefore
(1)P[X≥1]=∑i≥1(ni)(N−n⌊0.3N⌋−i)N⌊0.3N⌋,
where ⌊·⌋ is the greatest integer function. For our dataset, we need n≥19 to ensure that the right hand side of Equation (1) is at least 99%. Therefore, if a category has a frequency of 18 or lower, there is some chance that no single report from that category will be selected in the test data, making the classification task meaningless. To prevent this from happening, we removed the categories with a frequency of 18 or lower. Thus, we considered the following models and datasets:“Full data” with 1121 error reports and four broad error categories “Administrative”, “Treatment”, “Treatment Preparation” and “Standards”“Administrative data” with 363 error reports and two event categories“Treatment data” with 236 error reports and two event categories“Treatment Preparation data” with 260 error reports and three event categories“Standards data” with 228 error reports and two event categories.

### 3.1. Data Processing and Feature Extraction

We carried out the following data processing steps prior to fitting the classification model.

Text pre-processing: For both the training and test sets, the following standard NLP actions were performed:Punctuation removal: This involves removal of all punctuation symbols from the description of the error reports.Text normalization: This involves converting all the error reports to lowercase letters. This ensures that words represented as ‘DOCTOR’ and ‘doctor’, for example, are treated as the same word.Tokenization: This involves converting the text to tokens, which could be just words (unigrams) or might include *N*-grams (which are sequences of *N* words) for several choices of *N*. In our case, we took into consideration all possible unigrams, bigrams (sequences of two words), and trigrams (sequences of three words) from the error reports.Feature extraction and weighting: This step involves the construction of a numerical representation of the text documents. Corresponding to each report, we constructed a feature vector consisting of weights computed using a well-known technique called Term Frequency-Inverse Document Frequency (tf-idf) [17,18]. Tf-idf assigns a weight to a token in a document, with the token receiving more weight if it appears more frequently in the document, but it also inversely weights the token based on its frequency of appearance across all documents considered.Dimension reduction using Latent Semantic Analysis: We next applied Latent Semantic Analysis (LSA) [19,20] for dimension reduction of the tf-idf matrix. This involves performing a reduced rank Singular Value Decomposition (SVD) on the tf-idf matrix. We considered the singular vectors corresponding to the top 200 singular values for the full data and the top 50 singular values for the four other datasets.

### 3.2. Classification Procedure

After constructing the feature vectors, the next step is carry out classification. We considered the following machine learning algorithms for this step:Linear Support Vector Machine: Linear Support Vector Machine (Linear SVM) [21,22] is a supervised learning technique with well-known generalization properties, high accuracy, and computational efficiency. Since our data set is not very large with n=1121 error reports, the feature vectors are likely to be longer than the number of data points. Linear SVM is well-suited for such higher dimensional problems because it automatically applies regularization.Multilayer Perceptron (MLP): A multilayer perceptron is a type of artificial neural network that is fully connected [23]. It consists of an input layer, one or more hidden layers, and an output layer. The data were received by the input layer and transferred to the first hidden layer. Following that, these inputs were transformed using a weight and a bias and fed into a function known as the ‘activation function’. The resulting value is sent to the output layer or, if there is one, to the next hidden layer, and so on. The output layer is also associated with an activation function, which produces the desired output. To employ MLP, we performed a one-hot encoding of our label variable. For the full model, for example, the four broad categories are assigned values [1,0,0,0], [0,1,0,0], [0,0,1,0], and [0,0,0,1], and the data were then fed to the neural network. We considered one hidden layer of dimension 50 for each of the data sets.Convolutional Neural Network (CNN): Convolutional Neural Networks are a type of artificial neural network that is widely used in image classification. A CNN, like regular artificial neural networks, is made up of fully connected layers that are linked to activations in the previous layer. We chose this model based on previous research that has shown that CNNs can successfully classify sentences and medical text [24,25]. As with MLP, we considered 1 hidden layer with dimension 50 for this model. We consider 128 output filters for the convolution layer, with the length of the convolution window set to 5. These convolution layer hyperparameter values were determined through cross-validation using the following options: Number of output filters: 32, 64, 128; Convolution window length: 5, 7, 9.

We employed the following Python libraries for this analysis: the Linear SVM model was implemented using scikit-learn, the neural network based models were implemented using Keras, and the text processing was carried out using Spacy.

## 4. Results

After training the model, we investigated the model performance based on the test data. For the *i*th label in the data set, let $TP_{i}$, $TN_{i}$, $FP_{i}$, $FN_{i}$ denote the number of True Positives, True Negatives, False Positives, and False Negatives, respectively. For each label, we calculated the following performance metrics:$Precision_{i}=\frac{TP_{i}}{TP_{i}+FP_{i}}$$Recall_{i}=\frac{TP_{i}}{TP_{i}+FN_{i}}$F1-$score_{i}=\frac{2\times Precision_{i}\times Recall_{i}}{Precision_{i}+Recall_{i}}$

Let $n_{i}$ denote the true number of error reports in the test data with label *i*, and let $n_{c}$ denote the number of classification labels. From Figure 2 we know that the classes are highly imbalanced. Therefore, we used the following metrics to evaluate the model performance, which take into account the true size of each classification label.



$Balanced\;Accuracy=\frac{\sum_{i}Recall_{i}}{n_{c}}$



$Weighted\;Precision=\frac{\sum_{i}n_{i}Precision_{i}}{\sum_{i}n_{i}}$



$Weighted\;Recall=\frac{\sum_{i}n_{i}Recall_{i}}{\sum_{i}n_{i}}$

WeightedF1-$score=\frac{\sum_{i}n_{i}F1-score_{i}}{\sum_{i}n_{i}}$

### 4.1. Model Performance

In Table 1, Table 2, Table 3 and Table 4, we report the performance metrics for the following models, respectively. The performance metrics for each model are averaged across 10 replications, and the corresponding standard deviations are given in parentheses.

Tf-idf (for feature extraction) + Linear SVM (for classification)Tf-idf followed by LSA (for feature extraction) + Linear SVM (for classification)Tf-idf followed by LSA (for feature extraction) + MLP (for classification)Tf-idf followed by LSA (for feature extraction) + CNN (for classification)

Given the significant class imbalance, the precision value for neural network-based models may become undefined (Table 3 and Table 4). To address this, we used class weights, assigning higher weights to minority classes and lower weights to majority classes.

The automated labeling algorithms, according to these tables, performed quite accurately for the first three models, with performance metrics ranging from 90% to 100% in most cases. The algorithms are the most accurate for the Administrative data, followed by the Standards data, then the Treatment data, the Treatment Preparation data, and finally our Full data (considering all the metrics, and for all classification methods). This was expected because administrative errors were heavily over-represented in the training data, implying that our full model would learn more about the language from administrative errors than from any other type of error. However, the downside to this is that the full data-based classification procedure is also more biased towards classifying text as an administrative error. By comparing the results in Table 1 and Table 2, we can conclude that the model performance with and without LSA is comparable, indicating that the extra step of LSA is not required.

Looking at the results for the neural network-based models in Table 3 and Table 4, we see that the MLP-based model performs similarly to the SVM-based models, while the CNN-based model performs worse. This contradicts previous research, which shows that the CNN model performs well in similar tasks such as sentence categorization and medical text classification [24,25]. A possible explanation is that our data sets are too small, causing backpropagation to fail for CNN. It is also possible that the pooling layer is causing us to lose valuable information during the dimension reduction process.

### 4.2. Analyzing Reports Where Predicted Category Is Different from Reported Category

For a deeper dive into the model results, we examined two incidents in which the reported category differed from the predicted category. To ensure statistical robustness, we ran our second classification model (*Tf*-*idf* + LSA + SVM) 1000 times to consider the full distribution of predicted categories. The results are displayed in Table 5. Reporters labeled both entries as ‘Treatment’ errors, but the content of each entry suggests that the error occurred during the ‘Treatment Preparation’ phase. The model correctly predicts the broad category of ‘Treatment Preparation’ for the first incident 100% of the time and 88.9% of the time for the second incident.

We believe that the reporters may have mislabeled these entries in the MERP program. The MERP program prompts the reporter to indicate whether the error occurred before or after the start of patient treatment (pre-treatment error or post-treatment error). Patient setup instructions are typically entered in the pre-treatment phase of the plan (treatment preparation) at various points in the process tree such as CT simulation, treatment planning, initial patient setup (positioning) on the treatment machine, etc. These instructions include information such as field name, target-source distance, shift amount and directions, etc. A review of the accuracy of the text features in entries using “setup” showed the vast majority of actual errors occurred at the pre-treatment stage and not the post-treatment stage. These reporter entry mislabelings most likely occurred because of confusion as to where to enter the error in the MERP program. This is because new patient setup instructions may also be created in the post-treatment phase of subsequent treatments due to changes in patient anatomy, positioning, original shifts from isocenter or other marks, etc. Confusion over where to enter these types of errors in the MERP program may have contributed to the human error in these cases. These are two instances where our model, or future models, developed in a similar manner and were used as a background observer to error reporting, which would be useful to the workflow of staff members. Team members may be biased to report an error based on their current context, but the model assistance would detect the error and suggest a change in cases where a discrepancy between preparation language and active language is detected.

## 5. Discussion

The field of radiation oncology is highly vulnerable to errors due to complex and intrinsically interdependent treatment workflows. Ineffective error management can lead to reduced quality, increased inefficiency, and increased legal and/or regulatory liabilities. The complexity of the entire system connected with large numbers of interactions and inherent trust of digital transfers may result in errors not being identified. However, various periodic checks in siloed settings are often performed manually using paper and spreadsheets.

This study shows that NLP-aided statistical algorithms can perform automated error labeling with high accuracy, even with a relatively small training dataset. Note that many of these error reports are not ideal training material because they contain grammatical errors, incomplete language, acronyms, and medical jargon. Nonetheless, the models were able to, for the most part, correctly distinguish labels in both broad categories and event categories. Furthermore, the model was able to detect certain instances where the human reporter made a mistake in classifying the error, and the model was able to correctly identify the “true" root cause of the error (see Table 5). When the model failed, this limitation was most likely explained by the uneven distribution of the training data set rather than any issues with the model construction. The second implication of this study for future research, therefore, is that there is a clear path for improving the model. Moving forward, it will be critical to use evenly distributed and sufficiently extensive data sets to develop models that improve on the process outlined in this study. This will require the collection of rich data sets from various treatment centers to ensure robust training of the classification models. In this context, we would like to point out related recent research using an Explainable Artificial Intelligence (XAI) framework, which aims at shortening the distance between AI and clinical practice for the investigation of breast cancer invasive disease events [26].

Because of their ability to look past contextual bias and the heuristics of individual reporters and observe the larger patterns, these models have significant implications for the improvement of workflow and error reporting in radiation oncology. We can see this from the analysis where the model predictions differed from the reported labels. Our full model was able to notice nuanced differences in setup vs. active language and, as a result, was able to identify errors that, while labeled as ‘treatment’ errors by reporters, likely had their root cause in ‘treatment preparation’. While we do not advocate replacing human reporters with artificial intelligence that automatically detects and reports deviations, we do believe that the future of these models will involve the use of artificial intelligence “helpers”. These helpers will passively observe the actions of human reporters and make suggestions when these actions deviate from larger patterns, similar to a spell checker software. This type of AI implementation would keep control in the hands of human reporters while assisting them by providing access to experiences beyond the immediate reporter perspective. With time, an AI introduced in this manner would also learn more, improving its ability to make accurate predictions. Introducing this type of helper AI would require reporters to spend less time determining what the “correct” error classification is, thereby allowing them more time to treat and direct their attention to their patients. This will also reduce the number of different reports that share the same error classification in an institution’s or treatment center’s system, allowing for a more clear path for determining causal factors and initiating root cause analysis, and eventually, error mitigation or elimination.

We envision a future where predictive models such as the one we have developed will be embedded into automated detection or advisory tools for the entire radiation team [27]. Using models in this way allows the team members in charge of reporting and analyzing errors to retain agency while receiving guidance and access to larger trends. Furthermore, our models simulate a truly objective and quantifiable risk management strategy free from human subjectivity. This important benefit can help pivot the perspective on medical errors from a blame culture to a safety culture [28]. As we will discuss in Section 5, we propose that models like this one be incorporated into the overall treatment process as guides, giving practitioners more time to treat patients and relieving the entire radiation team from their reliance on individual heuristics.

## 6. Conclusions

Our research shows that even with a relatively small and noisy training database, NLP-based statistical algorithms show excellent accuracy in carrying out automated error labeling from free text reports. Our algorithms achieve performance metrics ranging from 90% to 100% in most cases. Remarkably, in some scenarios, the model was able to detect cases where the human reporter made a mistake in classifying the error, and in these cases, the model was able to correctly identify the “true” root cause of the error.

Overall, we believe that the results of this study are promising to the potential of AI utilization in the field of radiation oncology, and that they also offer several direct pathways for subsequent research. Our models provide the groundwork for a systematic understanding of the probability of an error occurring in the overall treatment process. The use of AI as a risk management tool could measurably improve patient safety, compliance, and efficiency in the radiotherapy industry. Prospective identification of incidents could assist in effective mitigation of unforeseen hazards. In conclusion, let us leverage the strengths of machine learning so that radiation oncologists and members of the entire treatment team are given more time and bandwidth to focus on their strengths and provide better care to patients. Most importantly, the manner in which error is identified, investigated, and corrected in such an automated labeling system can alleviate the fear of error and promote a no-fault, just culture [28].

## Figures and Tables

**Figure 1 diagnostics-13-01215-f001:**
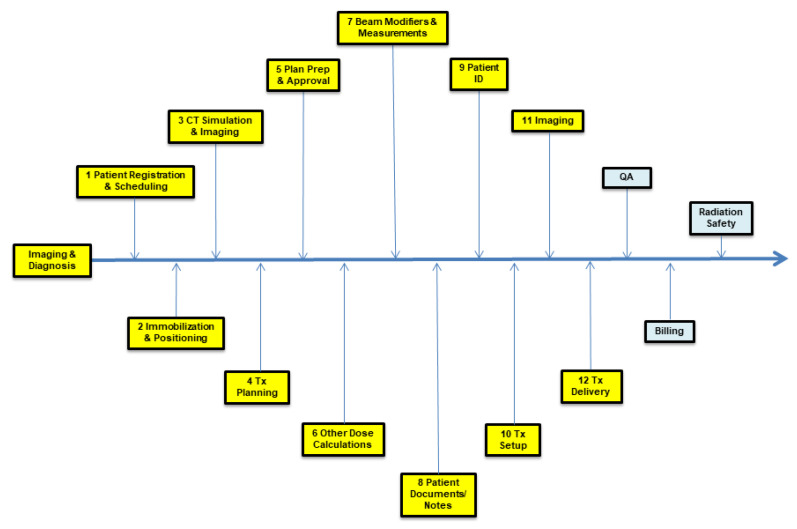
Process map showing the major steps in the radiation oncology workflow.

**Figure 2 diagnostics-13-01215-f002:**
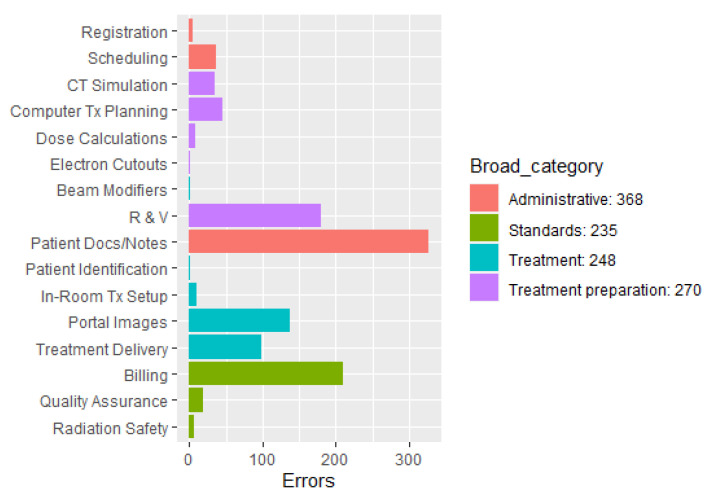
Frequency distribution of the 16 event categories (**left**) and the 4 broad categories (**right**).

**Table 1 diagnostics-13-01215-t001:** Model performance on test data with tf-idf + Linear SVM.

Metric	Full	Administrative	Treatment	Treatment Preparation	Standards
Balanced Accuracy	90.8 (1.1)	97.4 (2.8)	98.2 (1.1)	84.5 (5.3)	90.2 (5.0)
Weighted Precision	91.7 (0.8)	98.9 (0.9)	98.2 (1.1)	91.6 (3.3)	98.4 (0.8)
Weighted Recall	91.2 (0.9)	98.8 (1.1)	98.2 (1.2)	91.3 (3.2)	98.4 (0.8)
Weighted F1	91.2 (1.0)	98.8 (1.1)	98.2 (1.2)	90.9 (3.4)	98.3 (0.9)

**Table 2 diagnostics-13-01215-t002:** Model performance on test data with tf-idf + LSA + Linear SVM.

Metric	Full	Administrative	Treatment	Treatment Preparation	Standards
Balanced Accuracy	90.2 (1.6)	96.8 (3.4)	97.9 (1.1)	83.1 (5.4)	91.2 (5.9)
Weighted Precision	91.2 (1.3)	98.6 (0.9)	97.8 (1.2)	91.6 (3.1)	98.6 (0.9)
Weighted Recall	90.5 (1.5)	98.4 (1.2)	97.7 (1.2)	91.2 (3.0)	98.6 (0.9)
Weighted F1	90.6 (1.4)	98.5 (1.1)	97.8 (1.2)	90.7 (3.2)	98.4 (1.1)

**Table 3 diagnostics-13-01215-t003:** Model performance on test data with tf-idf + LSA + MLP.

Metric	Full	Administrative	Treatment	Treatment Preparation	Standards
Balanced Accuracy	88.9 (1.1)	97.2 (1.7)	97.7 (1.2)	89.4 (4.9)	99.0 (3.2)
Weighted Precision	90.3 (0.7)	97.3 (1.0)	97.6 (1.2)	90.1 (3.8)	99.9 (0.5)
Weighted Recall	89.2 (0.9)	96.4 (1.6)	97.5 (1.3)	86.7 (4.4)	99.9 (0.5)
Weighted F1	89.3 (0.9)	96.6 (1.5)	97.5 (1.3)	87.4 (4.1)	99.8 (0.5)

**Table 4 diagnostics-13-01215-t004:** Model performance on test data with tf-idf + LSA + CNN.

Metric	Full	Administrative	Treatment	Treatment Preparation	Standards
Balanced Accuracy	62.4 (4.6)	89.4 (2.9)	90.1 (3.9)	61.2 (5.9)	84.3 (7.5)
Weighted Precision	71.7 (6.1)	93.4 (1.2)	91.4 (3.6)	77.4 (4.4)	94.2 (1.8)
Weighted Recall	63.6 (4.6)	81.1 (5.2)	90.1 (4.0)	56.8 (7.3)	79.9 (17.3)
Weighted F1	63.8 (4.7)	84.5 (4.0)	90.2 (3.9)	60.1 (6.8)	83.4 (13.8)

**Table 5 diagnostics-13-01215-t005:** Predicted categories.

Error Report Text Description	Administrative	Treatment	Treatment Preparation	Standards
Patient’s LMT setup instructions indicated that the RMT TSD was changed. Incorrect naming of field in setup instrucitons.	0%	0%	100%	0%
Field #1 setup instructions state shift isocenter, but does not say in which direction.	0%	11.1%	88.9%	0%

## Data Availability

The data is unavailable due to proprietary ownership.

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
