# Peer review of "Automated Error Labeling in Radiation Oncology via Statistical Natural Language Processing"

_diagnostics, 2023, doi:10.3390/diagnostics13071215_

Round 1
Reviewer 1 Report
Abstract: IOM at first appearance should be expanded to Institute of Medicine
Remove citation from abstract, No citation at abstract section
You can mentioned the risk of error expected at oncology field (short sentence).
Keywords accepted in the current form.
Introduction: Please arrange the references number in this section well, (A milestone report from the Institute of Medicine (IOM), published in 2000, brought public and political attention to the severe fallout of medical errors and highlighted the need to address medical errors and their effects on human health [1]. More recently, medical 16 errors have been shown to be the third leading cause of death in the United States [5].) reference No 1 should be followed by reference No 2 not reference No 5 as appeared in the text.
Introduction and problem statement are too long compared to the discussion section can you compromised between two sections!
Materials and Methods: This is the clear section and attractive section in the manuscript.
Results also clear to me
Discussion need to be improved.
Conclusion: try to conclude the main findings through your results as possible.
Reviewer 2 Report
see the report
